# Regression shrinkage and selection via least quantile shrinkage and selection operator

**Alireza Daneshvar**, **Golalizadeh Mousa** *

Department of Statistics, Tarbiat Modares University, Tehran, Iran

☯ These authors contributed equally to this work.
* golalizadeh@modares.ac.ir

**Data Availability Statement:** https://cran.r-project.org/web/packages/flare/index.html.

**Funding:** The author(s) received no specific funding for this work.

## Abstract

Over recent years, the state-of-the-art lasso and adaptive lasso have aquired remarkable consideration. Unlike the lasso technique, adaptive lasso welcomes the variables' effects in penalty meanwhile specifying adaptive weights to penalize coefficients in a different manner. However, if the initial values presumed for the coefficients are less than one, the corresponding weights would be relatively large, leading to an increase in bias. To dominate such an impediment, a new class of weighted lasso will be introduced that employs all aspects of data. That is to say, signs and magnitudes of the initial coefficients will be taken into account simultaneously for proposing appropriate weights. To allocate a particular form to the suggested penalty, the new method will be nominated as 'lqsso', standing for the least quantile shrinkage and selection operator. In this paper, we demonstate that lqsso encompasses the oracle properties under certain mild conditions and delineate an efficient algorithm for the computation purpose. Simulation studies reveal the predominance of our proposed methodology when compared with other lasso methods from various aspects, particularly in ultra high-dimensional condition. Application of the proposed method is further underlined with real-world problem based on the rat eye dataset.

## Introduction

Most data analysts intend to accomplish two central goals while dealing with regression models in a high dimensional framework. The first aim is to profit high prediction accuracy. Another one which is recognized as interpretability, applies to the act of selecting pertinent explanatory variables that have an intense relationship with the response variable [1]. In other words, prediction accuracy refers to adjusting bias and variance components. It is important to note that managing bias and variance trade-off results in high prediction accuracy, provided that appropriate modeling methods have been designated. Regularization technique is a common strategy to design a convenient balance between these two quantities. With a convex penalty class, some regularization methods involve nonnegative garotte [2], ridge [3], lasso [4] and elastic net [5]. In association with the aforementioned tools, lasso has been greatly appreciated as a result of its statistical and applied properties. It is noteworthy to mention that adaptive lasso, defined by Zou [6] is a particular version of lasso that allocates adaptive weights to

**Competing interests:** The authors have declared
that no competing interests exist.

different coefficients by including some impressive properties in the $L_1$ penalty. As a result of
selecting a right subset model under some particular conditions, this method covers two afore-
mentioned properties, i.e., high prediction accuracy and sensible interpretability. It is worthy
to note that Tibshirani [7] has reviewed various statistical methods following out lasso.

It is common knowledge that lasso disregards the effect of random variables in the penalty
term. On the contrary, the adaptive lasso, suppresses such a drawback and as a result, has a bet-
ter precision in the statistical manner. This technique encompasses oracle properties, con-
forming the same algorithm as lasso does. In the mathematical context, adaptive weights are
determined using the initial estimates, derived via invoking OLS method for the estimation of
regression coefficients. The corresponding weights lead to high precision, improving the adap-
tive lasso to be authentic. Bühlmann and Van De Geer in [8] have mentioned the condition
that if initial coefficients are large, adaptive lasso employs a small penalty leading to little
shrinkage and less bias. In addition, neither of Zou in [6], Bühlmann and Van De Geer in [8]
and Fan and his colleagues in [9] have investigated a specific situation by which the absolute of
initial coefficients are less than one, resulting in extra bias. They have confined themselves to
either taking the zero coefficient as the initials or restricting the weights in some pre-deter-
mined bounds. However, if the absolute of initial coefficient is less than one, the corresponding
weight turns to be large. Indeed, in a regular regression (OLS), there is no penalty part, bias is
low and variance is high. By applying a penalty, we sacrifice bias (i.e. increase bias) to reduce
variance, considered as a bias-variance tradeoff. As a consequence, bias increases by adding a
penalty term. Based upon the reseach by Bühlmann and Van De Geer in [8], if the absolute of
the jth initial coefficients is large, the adaptive lasso enforces a minor penalty (i.e. little shrink-
age) for the jth coefficient, causing less bias. Therefore, our proposed method can exert the
mentioned subject. Specifically noted, our suggested method can overcome this problem by
supporting unbiased estimates in a situation by which the weights are greater than one. It
should be emphasized that adaptive lasso merely uses the magnitude of initial estimates. To
handle this issue, we propose a new method with the gurantee to provide weights between zero
and one, improving the accuracy of the new lasso family estimators better in comparison with
other common methods. As shall be seen, our proposed method takes the signs and magni-
tudes of the initial coefficients into account. These two interesting features of the new method
make it superior when compared with some current methods in modeling high dimensional
data. One of the main proof for our proposed method is Bayesian approach. Our proposed
method can be viewed through a Bayesian methodology. The lqsso can be considered as a
Maximum A Posteriori (MAP) estimator. This is the case when one takes the prior distribu-
tions for the initial coefficients as the Asymmetric Laplace Distribution (ALD) written by
Koenker and Machado in [10].

Specifically, we present a new weighted lasso, as an alternative to adaptive lasso for the esti-
mation of regression parameters in various situations. Our adaptive weights have been
inspired from quantile regression methodology, proposed by Koenker and Bassett in [11]. In
contrast with [12, 13], applying check function as a loss with an adaptive lasso penalty, in this
article we employ a common quadratic loss along with check function as a penalty to regularize
the associated parameters.

In analogy with titles appeared in the terminology of lasso, we call our proposed method
`lqsso`. We demonstrate that lqsso is a developed version of lasso and adaptive lasso. More-
over, we enforce that lqsso performs very well in comparison with other stated penalization
methods. We also illustrate that the suggested method covers oracle properties, the concept
advocated in [14]. Besides, lqsso is a convex optimization problem, like lasso and its exten-
sions, therefore it does not suffer from local minimum drawback. The procedure to implement
our proposed method is implicitly the same as lasso and adaptive lasso, accordingly we can

apply efficient algorithms from both procedures. However if there is concern about computational costs of implemented methods, one can use coordinate descent [15] and LARS [16] to derive estimates from the lqsso method.

The rest of this article is organized as follows. We first define our proposed penalization method, i.e. lqsso. Further, we present the algorithm and geometrical aspects of our method. As mentioned previously, our proposed method takes advantages of oracle properties under certain conditions. The essentials to exhibit this advantage are presented. We conduct simulation studies to compare lasso, adaptive lasso and lqsso and bring forward the results. We provide an application of real data analysis to display the estimation and variable selection performance of our method. In conclusion, a general discussion on lqsso and comprehension for future research are represented.

## Lqsso; a novel method of variable selection

### Definition

Suppose the pairs $(x_j, y)$, $j = 1, 2, \ldots, p$ are presented, where $y = (y_1, \ldots, y_n)^T$ and $x_j = (x_{1j}, \ldots, x_{nj})^T$ are the response and predictor variables, respectively. Also, let $X = [x_1, \ldots, x_p]$ be a predictor matrix. In the present article, it is assumed that $y_i$s are conditionally independent given $x_{ij}$s, remarking that $x_{ij}$ are centered and scaled, therefore $\sum_i x_{ij} = 0$, $\sum_i x_{ij}^2 / n = 1$. Consider a general linear model structure $y = X\beta + \varepsilon$, where $\beta = (\beta_1, \ldots, \beta_p)^T$, $\varepsilon = (\varepsilon_1, \ldots, \varepsilon_n)^T$ and $\varepsilon_1, \ldots, \varepsilon_n$ hint at independent identically distributed (i.i.d) random errors with zero mean and variance $\sigma^2$. In the rest, a subscript is assigned for the purpose of determining the estimation of coefficient $\beta$ derived from a particular method. We expect such definitions are self-described without any ambiguity wherever they first appear in this and subsequent sections. Besides, a superscript analogous to $(n)$ is applied to show dependency of the estimator on the sample size, and to investigate the asymptotic behavior of the estimator. An example is $\hat{\beta}_{lqsso}^{(n)} = (\hat{\beta}_{(1,lqsso)}^{(n)}, \ldots, \hat{\beta}_{(p,lqsso)}^{(n)})^T$.

At first step, there is necessity to specify our proposed estimator according to a minimizing problem. In this connection, let us define an estimator, say $\hat{\beta}^{(n)}$, for the parameter $\beta$, as

$$\hat{\boldsymbol{\beta}}^{(n)} = \underset{\boldsymbol{\beta}}{argmin} \sum_{i=1}^{n} (y_i - \sum_{j=1}^{p} \beta_j x_{ij})^2 + \lambda_n \sum_{j=1}^{p} \rho_\tau(\beta_j), \tag{1}$$

where $\lambda_n \geq 0$ is a tuning parameter, $\tau$ is a fixed number chosen from $(0, 1)$ and

$$\rho_\tau(\beta_j) = [\tau - I(\beta_j < 0)]\beta_j = [(1 - \tau)I(\beta_j \leq 0) + \tau I(\beta_j > 0)]|\beta_j|. \tag{2}$$

Note that the estimator in Eq (1) depends on $\lambda_n$. For this reason, the superscript $(n)$ is located over $\hat{\beta}$ to obliquely emphasize the dependency. Moreover, as comprehended, we add $\rho_\tau(\beta_j)$ in Eq (1) to express the adapting manner of estimator with observation in our proposed model. Additionally, due to the structure of the check function it can be realized that the considered penalty is flexible with quantile $\tau$. This is in contrast with other penalties proposed previously by which the response's quantile did not play a part in the penalty. Without loss of generality, we assume a condition by which intercept has been eliminated from the regression model. If this is not the case, we can simply center the response, then one will eccounter with the variables having a zero mean.

To fit a model, an important point to note is that the corresponding coefficients are not known formerly. So, initially imposing some criteria on coefficients in order to choose relevant weights, does not make sense in real application. In particular, we intend to turn this fact

around before implementing Eq (1). To tackle this case, we suppose that one has already derived the OLS estimates of the coefficients before intending to determine the corresponding weights appeared in our proposed method. Therefore, it is recommended to define lqsso estimate, $(\hat{\boldsymbol{\beta}}_{lqsso}^{(n)})$, as

$$\hat{\boldsymbol{\beta}}_{lqsso}^{(n)} = \underset{\boldsymbol{\beta}}{argmin} \sum_{i=1}^{n}(y_i - \sum_{j=1}^{p}\beta_j x_{ij})^2 + \lambda_n \sum_{j=1}^{p}\rho_\tau^*(\beta_j), \tag{3}$$

where two fixed parameters are the same as those defined in Eq (1) and

$$\rho_\tau^*(\beta_j) = [\tau - I(\hat{\beta}_{j,OLS} < 0)]\beta_j = [(1-\tau)I(\hat{\beta}_{j,OLS} \leq 0) + \tau I(\hat{\beta}_{j,OLS} > 0)]|\beta_j|.$$

As perceived, the proposed lqsso functions similar to lasso, as a result of considering magnitude and sign of the estimated coefficients, applying OLS in the penalty term. Typically, the proposed penalty captures all accessible information in samples, therefore lqsso is expected to perform better than alternative competing methods in many small effect situations.

Our motivation to outline lqsso is in view of selecting variables and minimizing bias when initial coefficients for the corresponding weights in adaptive lasso are less than one. In this regard, the function employed in penalty term provides less weight in proportion to irrelevant coefficients, analogous to the process of treating usual and outlier variables differently in statistical analysis. Indeed, the idea is inspired from integrating two interesting methods proposed by Tibshirani and also Koenker and Bassett in [4, 11], respectively. It is commonly known that lasso minimizes

$$\sum_{i=1}^{n}(y_i - \sum_{j=1}^{p}\beta_j x_{ij})^2 + \lambda_n \sum_{j=1}^{p}|\beta_j|, \tag{4}$$

while the quantile regression modeling attempts to minimize the following expression

$$\sum_{i=1}^{n}\rho_\tau(y_i - \sum_{j=1}^{p}\beta_j x_{ij}). \tag{5}$$

But, as noticed formerly in Eq (3), we applied Koenker's loss function as a penalty function. Our proposed method is further similar to adaptive lasso, aside from assigning different weights. Consider the weighted lasso,

$$\underset{\boldsymbol{\beta}}{argmin} \sum_{i=1}^{n}(y_i - \sum_{j=1}^{p}\beta_j x_{ij})^2 + \lambda_n \sum_{j=1}^{p}w_j|\beta_j|, \tag{6}$$

where $\boldsymbol{w} = (w_1, \ldots, w_p)^T$ refers to a known weight vector.

Assume that $\hat{\boldsymbol{\beta}}$ is an estimator of $\boldsymbol{\beta}^*$, e.g. $\hat{\boldsymbol{\beta}}_{(OLS)}$, where $\boldsymbol{\beta}^* = (\beta_1^*, \ldots, \beta_p^*)^T$ is a true coefficient vector. Choose a $\gamma > 0$, and define the weight vector as $\hat{\boldsymbol{w}} = 1/|\hat{\boldsymbol{\beta}}|^\gamma$. The adaptive lasso estimates, say $\hat{\boldsymbol{\beta}}_{alasso}^{(n)}$, are then defined as

$$\hat{\boldsymbol{\beta}}_{alasso}^{(n)} = \underset{\boldsymbol{\beta}}{argmin} \sum_{i=1}^{n}(y_i - \sum_{j=1}^{p}\beta_j x_{ij})^2 + \lambda_n \sum_{j=1}^{p}\hat{w}_j|\beta_j|. \tag{7}$$

It is worth noting that, similar to Eqs (4) and (7), our objective in Eq (3) is also a convex optimization problem, and a global minimizer can thus be attained. Bearing in mind these stated similarities, one can conclude that lqsso is, in fact, a weighted lasso problem available in

[6]. On account of resemblance between our penalization method and previous versions of lasso, we can apply convenient algorithms for solving adaptive lasso and lasso in order to calculate the lqsso estimates. Computational details regarding to implementing lqsso are provided in subsection **The algorithm of lqsso**.

The proposed lqsso has a closed-form and many representations can be used with lqsso penalty in various circumstances, including an applicable algorithm, invoking in a geometrical field, recalling a Bayesian aspect, proving the Oracle property and calculating KKT conditions in closed form in the quantile regression. The noteworthy aspect of check function, by way of loss or penalty function, is that it can be considered in various structures. At this juncture, we summarize some of them. The equivalent expressions are as follows:

$$\rho_\tau(u) \quad = [\tau - I(u \le 0)]u = [(1-\tau)I(u \le 0) + \tau I(u > 0)]|u|$$
$$= u[\tau I(u > 0) - (1-\tau)I(u \le 0)] = \frac{|u| + (2\tau - 1)u}{2}.$$

Depending upon the case by which a relevant ingredient, smooth the intended mathematical consideration, one of the above equivalent representations of the check function will be taken into account. By way of illustration, the last expression will be invoked in order to evaluate KKT conditions.

## The geometry of lqsso

To present a visual sense of our proposed method, S1 Fig demonstrates a sketch of this objective for the case $p = 2$. In this condition, the loss function is assumed quadratic. The elliptical contour of this function is displayed by an ellipse, centered at the OLS estimates. The constraint regions, visualized with diamond in gray color, indicate the estimates derived from lasso. In addition, the polyhedrals are the spaces identified for the estimates granted by lqsso. As derived, the contour touches square in the lasso process and this issue sometimes occurs at a corner, corresponding to a zero coefficient. In lqsso structure, the contours touch polyhedrals as well, but with more flexibilities in such situations. For instance, varying $\tau$ in its domain allow an irregular constrained region (rather than a regular space) for the coefficients. In contrast, space of a regular polygon is the only area to seek the candidate parameters in the lasso case. It appears that lqsso provides more complete view and information corresponding to the constraint regions than lasso, indicating more comprehensive insight of lqsso compared with lasso. This is mainly due to the fact that absolute function is a special case of check function; regarding as penalties in both lasso and lqsso. As an instance, we arbitrarily set $\tau = .2$ and $\tau = .8$. Note that lqsso with $\tau = 0.5$ is equivalent to lasso in a graphical view point, though this does not occur in reality. The reason is that lqsso is approximately equivalent to lasso. Specifically, the main difference between the stated methods is that lqsso includes indicator function of OLS coefficients in the penalty component, a reason for distinction from lasso when $\tau = 0.5$.

The related signs exhibit the direction of polygon in figure and OLS coefficients in the penalty part. Strictly speaking, OLS coefficients are based on a density function f(y—X), which examines the mass not only at the middle of the density but also at their two tails, especially when f(.) is asymmetric. As a consequence, the suggested methodology is more pliable when compared with lasso and adaptive. For more explanation, plot C in Zou [6] is presented in S2 Fig for lqsso (black points). Note that Zou [6] indicated them by a line in his graph, however for a better illustration, we preferred to plot them as points. Regarding the $\tau$ value and its related sign, the degree of closeness of lqsso line to the red line as well as the positive or negative side can be achived. As formerly mentioned about Bayesian aspect, it should be noted that the value of $\tau$ is required to be fixed initially for calculations via invoking the cross-validation

technique. By assuming $\tau = 0.8$, with $\beta$ more than zero and less than one (consequently the related weight would be more than one), we are able to assign a probability for this true coefficient in conjunction with a reasonable probability for zero coefficients, i.e., sparsity. This notion relates to signs. Moreover, our calculated weights would be between zero and one, as yet. Note that the proposed method advocates the highest probability corresponding to zero for the adaptive case, as can be realized in S3 Fig.

## The algorithm of lqsso

The following section deals with computational aspects of implementing lqsso. Similar to alternative methods in the class of lasso, a coordinate descent algorithm can also provide the lqsso estimates after invoking Eq (3). This algorithm is available in the *glmnet* package [15], freely available in statistical software R. The *glmnet* package covers all the computational aspects related to the $L_1$ penalty and its corresponding extensions. Recalling discussion in the aforementioned sections, it is straightforward to delineate a procedure for implementing the suggested lqsso. This subject is briefly included in **Algorithm** 1.

**Algorithm** 1 (coordinate descent algorithm for deriving lqsso estimate):

```
1. Define x*ⱼ = xⱼ/wⱼ�q, where j = 1, 2, ..., p.
2. Solve the lasso problem for all λₙ, i.e.
```

$$\hat{\boldsymbol{\beta}}_{lasso}^{(n)} = \underset{\boldsymbol{\beta}}{argmin}\sum_{i=1}^{n}(y_i - \sum_{j=1}^{p}\boldsymbol{x}_j^*\beta_j)^2 + \lambda_n\sum_{j=1}^{p}|\beta_j|.$$

```
3. Output β̂⁽ⁿ⁾₍ⱼ,ₗqₛₛₒ₎ = β̂⁽ⁿ⁾₍ⱼ,ₗₐₛₛₒ₎/wⱼᵠ, j = 1, 2, ..., p.
```

At this point, we supply a brief sketch of a proof by which the **Algorithm** 1 guarantees a solution. At first, we consider the following equivalent expressions

$$\hat{\boldsymbol{\beta}}_{lqsso}^{(n)} = \underset{\boldsymbol{\beta}}{argmin}\sum_{i=1}^{n}(y_i - \sum_{j=1}^{p}\beta_j x_{ij})^2 + \lambda_n\sum_{j=1}^{p}w_j^q|\beta_j|$$

$$= \underset{\boldsymbol{\beta}}{argmin}\sum_{i=1}^{n}\left(y_i - \sum_{j=1}^{p}\beta_j x_{ij}\frac{w_j^q}{w_j^q}\right)^2 + \lambda_n\sum_{j=1}^{p}w_j^q|\beta_j|$$

$$= \underset{\boldsymbol{\beta}}{argmin}\sum_{i=1}^{n}(y_i - \sum_{j=1}^{p}\beta_j w_j^q x_{ij}^*)^2 + \lambda_n\sum_{j=1}^{p}w_j^q|\beta_j|$$

$$= \underset{\boldsymbol{\beta}^\alpha}{argmin}\sum_{i=1}^{n}(y_i - \sum_{j=1}^{p}\beta_j^\alpha x_{ij}^*)^2 + \lambda_n\sum_{j=1}^{p}|\beta_j^\alpha|,$$

where $\beta_j^\alpha = \beta_j w_j^q$. Implementing the aforementioned algorithm on the last expression gives $\hat{\beta}_{(j,lasso)}^{\alpha(n)} = w_j^q \hat{\beta}_{(j,lqsso)}^{(n)}$. It is then straightforward to recognize that $\hat{\beta}_{(j,lqsso)}^{(n)} = \frac{\hat{\beta}_{(j,lasso)}^{\alpha(n)}}{w_j^q}$.

Determining regularization parameter is a significant stage in all penalized regression problems. Customarily, we employ $\hat{\boldsymbol{\beta}}_{(OLS)}$ to compute the related weights in lqsso. But, following Zou [6], we can use $\hat{\boldsymbol{\beta}}_{(ridge)}$ instead of $\hat{\boldsymbol{\beta}}_{(OLS)}$, in high-dimensional case. Then, the objective is to obtain the optimal pairs of $(\tau, \lambda_n)$. This function is similar to the technique for applying adaptive lasso, where we intend to derive optimal pairs of $(\gamma, \lambda_n)$. According to Zou [6], the adaptive lasso uses cross-validation to tune these pair parameters. While implementing lqsso technique, we further utilize the same procedure to derive $(\tau, \lambda_n)$.

## Oracle properties of lqsso

This section provides oracle properties in the first phase. In the subsequent, we ascertain that our proposed penalization method (lqsso) follows the mentined features subject to some mild conditions. In particular, the subsequent **Theorem** demonstrates that lqsso covers oracle properties provided that a proper $\lambda_n$ is selected.

Let $\mathbb{A} = \{j : \beta_j^* \neq 0\}$ where $\beta_j^*$ is a $j$-th true coefficient and assume that the cardinality of $\mathbb{A}$ equals $p_0$, i.e. $|\mathbb{A}| = p_0$ such that $p_0 < p$. As a consequence, the true model depends only on a subset of covariates, having a strong relationship with response variables. Note that $\frac{1}{n}X^TX = C$ where $C$ is a positive definite matrix. Generally speaking, the estimated regression coefficients, $\hat{\beta}_1, \ldots, \hat{\beta}_p$, possess the oracle properties, defined by Fan and Li in [14], if they satisfy the following conditions:

- They present a true subset model, i.e. $\{j : \hat{\beta}_j \neq 0\} = \mathbb{A}$.

- They follow asymptotic normality, i.e. $\sqrt{n}(\hat{\boldsymbol{\beta}}_{\mathbb{A}} - \boldsymbol{\beta}_{\mathbb{A}}^*) \xrightarrow{d} N(\mathbf{0}_p, \boldsymbol{\Sigma})$, where $\Sigma$ refers to the covariance matrix of the true subset model.

**Theorem**[Oracle properties] Assume $\frac{\lambda_n}{\sqrt{n}} \to 0$ and $\sqrt{n}\lambda_n \to \infty$. Then, the lqsso estimates ought to have the following properties:

- Sparsity: $lim_n P(\mathbb{A}_n^{lqsso} = \mathbb{A}) = 1$ as $n \to \infty$, where $\mathbb{A}_n^{lqsso} = \{j : \hat{\beta}_{\{j,lqsso\}}^{(n)} \neq 0\}$.

- Asymptotic normality: $\sqrt{n}(\hat{\beta}_{\{\mathbb{A},lqsso\}}^{(n)} - \beta_{\mathbb{A}}^*) \xrightarrow{d} N(\mathbf{0}_{p_0}, \sigma^2 \times C_{11}^{-1})$,

where $C_{11}$ is a $p_0 \times p_0$ matrix; a component of $C$ partitioned as:

$$C = \begin{bmatrix} C_{11} & C_{12} \\ C_{21} & C_{22} \end{bmatrix}.$$

Proof of **Theorem**:

At first, the asymptotic normality proof of the estimator derived from our proposed method, i.e., lqsso will be presented.

Let us consider $\boldsymbol{\beta} = \boldsymbol{\beta}^* + \frac{u}{\sqrt{n}}$ and

$$\Psi_n(\boldsymbol{u}) = ||\boldsymbol{y} - \sum_{j=1}^p \boldsymbol{x}_j\left(\beta_j^* + \frac{u_j}{\sqrt{n}}\right)||^2 + \lambda_n \sum_{j=1}^p w_j^q |\beta_j^* + \frac{u_j}{\sqrt{n}}|.$$

If $\hat{\boldsymbol{u}}_{(n)} = argmin_{\boldsymbol{u}} \Psi_n(\boldsymbol{u})$; then $\beta_{lqsso}^{*(n)} = \boldsymbol{\beta}^* + \frac{\hat{\boldsymbol{u}}_{(n)}}{\sqrt{n}}$ or $\hat{\boldsymbol{u}}_{(n)} = \sqrt{n}(\boldsymbol{\beta}_{lqsso}^{*(n)} - \boldsymbol{\beta}^*)$. Note that $\Psi_n(\boldsymbol{u}) - \Psi_n(\mathbf{0}) = V_4^{(n)}(\boldsymbol{u})$, where

$$V_4^{(n)}(\boldsymbol{u}) \equiv \boldsymbol{u}^T\left(\frac{1}{n}X^TX\right)\boldsymbol{u} - 2\frac{\boldsymbol{\varepsilon}^TX}{\sqrt{n}}\boldsymbol{u} + \frac{\lambda_n}{\sqrt{n}}\sum_{j=1}^p w_j^q\sqrt{n}(|\beta_j^* + \frac{u_j}{\sqrt{n}}| - |\beta_j^*|).$$

We know that $\frac{1}{n}X^TX \to C$ and $\frac{\boldsymbol{\varepsilon}^TX}{\sqrt{n}} \xrightarrow{d} W = N(\mathbf{0}_p, \sigma^2 C)$. Now consider the limiting behavior of the third term appeared in $V_4^{(n)}(\boldsymbol{u})$. Note that our weights include $\tau$, $1 - \tau$ and an indicator function of the initial coefficients, i.e. $I(\hat{\beta}_{(j,OLS)} > 0)$ and $I(\hat{\beta}_{(j,OLS)} \leq 0)$. Also, we know that the indicator functions converge, in probability, to an indicator function. Next, we consider three distinct cases regarding the value of $\beta_j^*$.

If $\beta_j^* > 0$, then $\sqrt{n}(|\beta_j^* + \frac{u_j}{\sqrt{n}}| - |\beta_j^*|) \rightarrow u_j sgn(\beta_j^*)$. Consequently, we have $\frac{\lambda_n}{\sqrt{n}}\tau\sqrt{n}(|\beta_j^* + \frac{u_j}{\sqrt{n}}| - |\beta_j^*|) \overset{P}{\rightarrow} 0$, following the Slutsky's theorem.

If $\beta_j^* = 0$, then $\sqrt{n}(|\beta_j^* + \frac{u_j}{\sqrt{n}}| - |\beta_j^*|) \rightarrow u_j$. Thus, we have $\frac{\lambda_n}{\sqrt{n}}(1-\tau)\sqrt{n}(|\beta_j^* + \frac{u_j}{\sqrt{n}}| - |\beta_j^*|) \overset{P}{\rightarrow} 0$ by, again, using the Slutsky's theorem.

If $\beta_j^* < 0$, then, $\sqrt{n}(|\beta_j^* + \frac{u_j}{\sqrt{n}}| - |\beta_j^*|) \rightarrow u_j sgn(\beta_j^*)$. Consequently, we have $\frac{\lambda_n}{\sqrt{n}}(1-\tau)\sqrt{n}(|\beta_j^* + \frac{u_j}{\sqrt{n}}| - |\beta_j^*|) \overset{P}{\rightarrow} 0$, by invoking the Slutsky's theorem one more time.

So, we see that $V_4^{(n)}(\boldsymbol{u}) \overset{d}{\rightarrow} V_4(\boldsymbol{u})$ for every $\boldsymbol{u}$, where

$$V_4(\boldsymbol{u}) = \boldsymbol{u}_{\mathbb{A}}^T \boldsymbol{C}_{11} \boldsymbol{u}_{\mathbb{A}} - 2\boldsymbol{u}_{\mathbb{A}}^T \boldsymbol{W}_{\mathbb{A}} \quad \text{if} \quad u_j = 0, \quad \forall j \notin \mathbb{A}.$$

Note that function $V_4^{(n)}$ is convex, and the unique minimum of $V_4$ is $(\boldsymbol{C}_{11}^{-1}\boldsymbol{W}_{\mathbb{A}}, \boldsymbol{0})^T$.

Following the epi-convergence results reported in Geyer [17] and Fu and Knight [18], we have

$$\hat{\boldsymbol{u}}_{\mathbb{A}}^{(n)} \overset{d}{\rightarrow} \boldsymbol{C}_{11}^{-1}\boldsymbol{W}_{\mathbb{A}} \quad and \quad \hat{\boldsymbol{u}}_{\mathbb{A}^c}^{(n)} \overset{d}{\rightarrow} \boldsymbol{0}. \tag{8}$$

In conclusion, we notice that $\boldsymbol{W}_{\mathbb{A}} \sim N(\boldsymbol{0}_{p_0}, \sigma^2 \boldsymbol{C}_{11})$; which completes the asymptotic normality of lqsso.

At present, we reveal the consistency of lqsso. The asymptotic normality result indicates that $\hat{\beta}_{(j,lqsso)}^{(n)} \overset{P}{\rightarrow} \beta_j^*$ for $\forall j \in \mathbb{A}$; thus $P(j \in \mathbb{A}_n^{lqsso}) \rightarrow 1$. Then, it suffices to show that $\forall j' \notin \mathbb{A}$, $P(j' \in \mathbb{A}_n^{lqsso}) \rightarrow 0$.

Consider the event $j' \in \mathbb{A}_n^{lqsso}$. By the Karush-Kuhn-Tucker (KKT) optimality conditions (see, e.g. Bühlmann and Van De Geer [8]), we know that $2\boldsymbol{x}_{j'}^T(\boldsymbol{y} - \boldsymbol{X}\hat{\boldsymbol{\beta}}_{lqsso}^{(n)}) = \lambda_n w_{j'}^q$. Note that $\frac{\lambda_n}{\sqrt{n}}w_{j'}^q \overset{P}{\rightarrow} 0$ whereas

$$2\frac{\boldsymbol{x}_{j'}^T(\boldsymbol{y} - \boldsymbol{X}\hat{\boldsymbol{\beta}}_{lqsso}^{(n)})}{\sqrt{n}} = 2\frac{\boldsymbol{x}_{j'}^T\boldsymbol{X}\sqrt{n}(\boldsymbol{\beta}^* - \hat{\boldsymbol{\beta}}_{lqsso}^{(n)})}{n} + 2\frac{\boldsymbol{x}_{j'}^T\boldsymbol{\varepsilon}}{\sqrt{n}}.$$

By Eq (8) and the Slutsky's theorem, we know that the quantity $2\frac{\boldsymbol{x}_{j'}^T\boldsymbol{X}\sqrt{n}(\boldsymbol{\beta}^* - \hat{\boldsymbol{\beta}}_{lqsso}^{(n)})}{n}$ converges to a normal density in distribution and $2\frac{\boldsymbol{x}_{j'}^T\boldsymbol{\varepsilon}}{\sqrt{n}}d \rightarrow^d N(\boldsymbol{0}, 4||\boldsymbol{x}_{j'}||^2\sigma^2)$. Thus

$$P(j' \in \mathbb{A}_n^{lqsso}) \leq P(2\boldsymbol{x}_{j'}^T(\boldsymbol{y} - \boldsymbol{X}\hat{\boldsymbol{\beta}}_{lqsso}^{(n)}) = \lambda_n w_{j'}^q) \longrightarrow 0.$$

This proves the consistency of lqsso. Note that based upon the knowledge from elementary statistics, we applied a simple property for the last stage of convergence. The mentioned property states that the probability for the case by which a continuous variable, with a continuous distribution, equals to a constant value is zero. The important point to highlight is that, unlike our procedure, Zou [6] invoked a property by which normal distribution tends to zero at its tails, in other words corresponding variables tend to infinity.

Note that **oracle property**, along with implementing a simple adoption from $l_1$ penalty, insure that our proposed method follows oracle properties.

## Simulation studies

In this section, we present the results of simulation studies in order to illustrate the performance of our proposed method. Considering that our intend is to compare lqsso with lasso

and adaptive lasso, we pursue to maintain the same spirit of simulation scheme investigated by Zou [6]. Hence, we take into account the effects of the same parameters that he considered, i.e. $\frac{\sigma_x^2}{\sigma^2}$, $\sigma^2$ and $n$, respectively, indicating the Signal-to-Noise Ratio (SNR), the variance of the error and the sample size. Because we can write $E[(\hat{y} - y_{test})^2] = E[(\hat{y} - \boldsymbol{X}^T\boldsymbol{\beta})^2] + \sigma^2$, we report the Relative Prediction Error (RPE), defined as $RPE = \frac{E[(\hat{y} - \boldsymbol{X}^T\boldsymbol{\beta})^2]}{\sigma^2}$, for comparing different regression methods discussed in this paper combined with various scenarios. To conduct our simulation studies, we apply various linear models represented by $y = \boldsymbol{x}^T\boldsymbol{\beta} + N(0, \sigma^2)$ through altering sample size (n), SNR and the error variance ($\sigma^2$). As frequently, we take OLS coefficient estimates as the initials for weights when considering adaptive lasso and lqsso. In the high-dimensional setting, the ridge coefficient estimates are yet determind as initial coefficients for those weights. The coordinate descent algorithm proposed by Friedman et al. [15], available in the *glmnet* package is then implemented to compute estimates of the relevant parameters while fitting the three delibrated methods. For each method, we select $\lambda_n$ and $\gamma$ in adaptive lasso and $\tau$ for lqsso, using a set of conceivable values. The selected values for $\lambda_n$, $\gamma$ and $\tau$ are {0.1, 0.2, ..., 2}, {0.1, 0.2, ..., 2} and {0, 0.01, ..., 1}, respectively. Thereupon, the sum of squared difference between all estimated coefficients and their true values has been calculated in order to inspect bias. In this manner, the mean of 100 simulation runs has been reported as a measure of bias for each method. In subsequent, the outcomes of aforesaid models in connection with various scenarios assumed in our simulation studies are presented.

Low dimensional case

**Model 1**: We set $\boldsymbol{\beta} = (\beta_1, \beta_2, \beta_3, \beta_4, 0, 0, 0, 0)^T$, where $\beta_1, \beta_2, \beta_3, \beta_4$ are independently generated from standard normal distribution. The covariates $\boldsymbol{x}_i$ ($i = 1, ..., n$) are *i.i.d* random vectors generated from 8 dimensional standard multivariate normal distribution. To impose collinearity among variables, we define correlation between each pair of predictor variables, $x_{ij}$ and $x_{ij'}$, through the expression $cor(j, j') = (0.5)^{|j - j'|}$, $1 \leq j, j' \leq 8$. We also set $\sigma$ equal to 1, 3 and 6 where the corresponding SNRs are 21.25, 2.35 and 0.59, respectively. The sample size (n) is fixed at 40 and 80.

**Model 2**: This model is similar to **Model 1**, except all $\beta_j$ are independently generated from standard normal distribution for $j = 1, ..., 8$.

High dimensional setting

For two alternative models, we specify all parameters and simulation settings similar to **Model 1**, besides the numbers of variables which are set at $p = 100$.

**Model 3** (Dense): All 100 coefficients are independently generated from standard normal distribution.

**Model 4** (Sparse): A total of 30 non-zero coefficients are independently simulated from standard normal distribution.

Ultra high dimensional setting

At this point, we merely consider one model with number of variables ($p$) equal to 1000.

**Model 5** (Very sparse): Only 30 coefficients are non-zero. The assumed coefficients are independently generated from normal distribution with mean and standard deviation equal to 0.5. Remainder coefficients; from the total of 970, are set at zero.

In what follows, we are going to provide more details for the simulated samples and processing them for major analysis. In addition, we present the notification of results. To evaluate the considered models, standard procedures have been carried out. In consequence, simulated observations were divided into two sections: training data and test samples. The number of training samples were fixed at 100 for each scenario analyzed formerly. Extra 1000 samples were employed as test set. To derive appropriate values for $\lambda_n$, $\gamma$ and $\tau$, RPE criterion was implemented, while $n$ and $\sigma$ were set as discussed previously.

To evaluate the accuracy of RPE, the related standard errors were computed through a bootstrap scheme, in the following manner. A fixed sample of RPEs were generated considering as bootstrapped samples. Afterwards, the median of the bootstrapped samples were extracted. The procedure was thus repeated 500 times. Standard deviation of medians, denominating as Monte Carlo *sd*, was reported for the estimated standard error of RPEs.

To demonstrate various aspects of simulation studies and for the purpose of comparison, the outputs were intentionally divided into sections. In other words, we separated the results to low, high and ultra high-dimensional settings, similar to the structure defined in the simulation study. We then prepared the results equivalent to the separation process. It is necessary to mention that more extensive simulations can be supplied to underline other aspects of the proposed methodology. Nevertheless, to save space and to emphasize on more appropriate results, we preferred to focus on aspects highlighting the proposed method. Henceforth, we cover the outcomes of our investigations.

The results derived from simulation studies in the low-dimensional settings are demonstrated in Tables 1 and 2. Some essential remarks extracted from these two tables are as follows. Focusing on **Model 1**, it can be achieved that for different values of SNR, $n$ and $\sigma$, the adaptive lasso is performing better than lasso and lqsso in terms of RPE and bias criteria. From the perspective of RPE, the proposed method has a better performance than lasso. With some minor exceptions for lasso in terms of bias, lasso and lqsso had no superiority over each other. Pointed out that **Model 1** refers to low dimensional setting.

Under construction of **Model 2**, the proposed method, i.e. lqsso, performs better than two alternative methods in terms of RPE and bias. Interestingly, standard deviation of RPEs corresponding to our suggested methodology is also lower compared with the corresponding values of lasso and adaptive lasso. Also like to point out that **Model 2** regards to sparse situation in the lower dimensional setting. Hence, the lqsso is recognizing the sparsity better than two standard methods in the regression modeling framework. Additionally, **Model 2** reveals that to incorporate with high variability among data expressed in the weights for coefficients, lqsso uses the information available in data better than the adaptive lasso. Bear in mind that we

**Table 1. The mean values of RPEs for Model 1 and Model 2.**

| | | | Method | | |
|---|---|---|---|---|---|
| **Model** | $n$ | $\sigma$ | **lasso** | **alasso** | **lqsso** |
| **Model 1** | 40 | 1 | 0.334 (0.021) | **0.163** (0.019) | 0.297 (0.027) |
| | | 3 | 0.181 (0.014) | **0.126** (0.009) | 0.163 (0.015) |
| | | 6 | 0.137 (0.017) | **0.121** (0.010) | 0.123 (0.018) |
| | 80 | 1 | 0.206 (0.010) | **0.051** (0.004) | 0.203 (0.014) |
| | | 3 | 0.084 (0.007) | **0.065** (0.005) | 0.079 (0.006) |
| | | 6 | 0.075 (0.006) | **0.059** (0.006) | 0.074 (0.006) |
| **Model 2** | 40 | 1 | 0.432 (0.030) | 0.434 (0.032) | **0.345** (0.025) |
| | | 3 | 0.258 (0.010) | 0.262 (0.010) | **0.253** (0.011) |
| | | 6 | 0.220 (0.013) | 0.226 (0.021) | **0.210** (0.018) |
| | 80 | 1 | 0.314 (0.022) | 0.308 (0.022) | **0.242** (0.013) |
| | | 3 | 0.109 (0.008) | 0.110 (0.008) | **0.103** (0.005) |
| | | 6 | 0.116 (0.008) | 0.117 (0.008) | **0.113** (0.008) |

The mean values of RPEs and their standard errors (in bracket), after fitting **Model 1** and **Model 2** using lasso, adaptive lasso (alasso) and lqsso methods for the simulated data. For each scenario, the selected methods in each row are highlighted in bold.

**Table 2. The mean values of bias for Model 1 and Model 2.**

| Model | $n$ | $\sigma$ | Method: lasso | alasso | lqsso |
|---|---|---|---|---|---|
| Model 1 | 40 | 1 | 0.166 (0.077) | **0.104** (0.012) | 0.170 (0.064) |
| | | 3 | 0.342 (0.070) | **0.294** (0.006) | 0.346 (0.072) |
| | | 6 | 0.597 (0.045) | **0.551** (0.006) | 0.594 (0.047) |
| | 80 | 1 | 0.139 (0.063) | **0.059** (0.012) | 0.142 (0.080) |
| | | 3 | 0.245 (0.059) | **0.222** (0.006) | 0.234 (0.062) |
| | | 6 | 0.429 (0.058) | **0.396** (0.005) | 0.433 (0.064) |
| Model 2 | 40 | 1 | 0.194 (0.204) | 0.197 (0.021) | **0.165** (0.037) |
| | | 3 | 0.400 (0.047) | 0.400 (0.028) | **0.388** (0.032) |
| | | 6 | 0.800 (0.022) | 0.803 (0.009) | **0.765** (0.027) |
| | 80 | 1 | 0.169 (0.249) | 0.171 (0.042) | **0.137** (0.037) |
| | | 3 | 0.263 (0.042) | 0.267 (0.028) | **0.254** (0.033) |
| | | 6 | 0.540 (0.042) | 0.542 (0.025) | **0.539** (0.043) |

The mean values of bias and their standard errors (in bracket), after fitting **Model 1** and **Model 2** using lasso, adaptive lasso (alasso) and lqsso methods for the simulated data. For each scenario, the selected methods in each row are highlighted in bold.

don't intend to make a comparison between lasso and alasso. This is because of the fact that such comparison needs to be done precisely in terms of both mentioned criteria and the boot-strapped sd, it also requires invoking many debates related to the stated methods.

In general, regardless of the presumed model, the estimated standard errors for all methods tend to decrease by increasing the sample size. This conclusion is exactly the same as what we expect from the asymptotic behavior of estimators in the context of statistical inference. To verify the asymptotic behavior of the mentioned methods, we focus on more scientific details. In the low dimensional setting by which the sparsity also exists, we claim that our proposed methodology performs better compared with lasso and alasso. It should be noted that lqsso can be considered as an economical method in the context of having low bias, sparsity and trade-off between the bias and variance, simultaneously.

Based upon the results reported in Tables 3 and 4, at first instance it might be difficult to make an explicit decision in declaring the best method based on scenarios considered in **Model 3** and **Model 4**. However, we can assert that our proposed method outperforms two alternatives. Such a conclusion might be a little optimistic statement based on the values reported in Table 4. But, the results in Table 3 confirms that our method has the least RPE for all scenarios considered for the **Model 3** and **Model 4**. Our method only lost to the alasso in some cases in terms of bias as it is evident in Table 4. However, in the lost cases, the difference between values of bias obtained from lqsso and alasso is negligible, which might be as a result of some minor computation roundings.

But as highlighted in Tables 3 and 4, lqsso is superior to lasso for all scenarios. It should be noted that **Model 3** and **Model 4** demonstrate situations by which one experience with high dimensional data analysis in dense and sparse cases, respectively.

The situation is rather advantageous promising in the ultra high-dimensional setting. As demonstrated in Tables 3 and 4, it can be achieved that while invoking **Model 5**, our proposed method (lqsso) has the best performance in comparison with two competitive methods, i.e. lasso and adaptive lasso. Both tables indicate that our method detects sparsity very well and has low RPE and bias. Interestingly, the suggested method manage to retrieve the information

**Table 3. The mean values of RPEs for Model 3, Model 4 and Model 5.**

| | | | Method | | |
|---|---|---|---|---|---|
| Model | $n$ | $\sigma$ | lasso | alasso | lqsso |
| Model 3 | 40 | 1 | 1109.216 (1.628) | 1110.291 (1.706) | **964.131** (2.583) |
| | | 3 | 167.084 (0.884) | 161.840 (0.644) | **161.391** (1.241) |
| | | 6 | 40.717 (0.184) | 40.901 (0.164) | **40.300** (0.285) |
| | 80 | 1 | 391.573 (5.711) | 371.177 (4.848) | **240.280** (2.582) |
| | | 3 | 43.058 (0.416) | 44.073 (0.418) | **42.705** (0.552) |
| | | 6 | 13.532 (0.211) | 13.559 (0.211) | **13.508** (0.185) |
| Model 4 | 40 | 1 | 32.700 (0.377) | 33.031 (0.353) | **31.753** (0.570) |
| | | 3 | 8.200 (0.137) | 8.218 (0.127) | **8.111** (0.123) |
| | | 6 | 3.396 (0.082) | 3.343 (0.063) | **3.311** (0.072) |
| | 80 | 1 | 3.146 (0.081) | 3.340 (0.088) | **2.657** (0.072) |
| | | 3 | 2.370 (0.059) | 2.210 (0.081) | **2.002** (0.050) |
| | | 6 | 1.369 (0.030) | 1.357 (0.032) | **1.206** (0.035) |
| Model 5 | 40 | 1 | 226.060 (2.656) | 162.295 (2.094) | **98.148** (0.829) |
| | | 3 | 46.331 (0.487) | 14.542 (1.226) | **4.573** (0.086) |
| | | 6 | 9.213 (0.120) | 4.956 (0.148) | **3.211** (0.067) |
| | 80 | 1 | 28.020 (0.597) | 15.912 (0.462) | **8.673** (0.235) |
| | | 3 | 3.779 (0.128) | 3.254 (0.109) | **2.949** (0.065) |
| | | 6 | 3.577 (0.070) | 2.802 (0.074) | **1.680** (0.046) |

The mean values of RPEs and their standard errors (in bracket), after fitting **Model 3**, **Model 4** and **Model 5** using lasso, adaptive lasso (alasso) and lqsso methods. For each scenario, the selected methods in each row are highlighted in bold.

among data and employs them to choose feasible weights for coefficients. Based upon conclusions achieved from simulation setting, we claim that our proposed method is better than adaptive lasso and lasso in ultra high-dimensional variable selection setting.

Note that the results and remarks presented in this section, were all based upon some particular scenarios and simulation settings. In this section, inspite the fact that various aspects of regression modeling were covered, a general conclusion can not be accomplished. Such consideration has also been addressed by Zou [6], based upon his simulation studies. In this regard, he also intended to make a decision on preferring between adaptive lasso and lasso methods.

To prepare a graphical conception in terms of bias and RPE for comparing each method based on different components appeared in the modeling process, i.e. sample size ($n$), standard deviation of error ($\sigma$), and five aforementioned models, S4 and S5 Figs are presented. At present, we do not discuss the remarks acquired from each figure, because the related results have already been presented while discussing the outputs in the previous tables. As stated, an specific decision on selecting the best candidate method is not straightforward. Nonetheless, according to the presented figures, the proposed lqsso method did relatively well in most cases.

In conclusion, the proposed method performed well in most scenarios and particularly in ultra high-dimensional setting. As a result, we are interested in evaluating its performance in more details. In this manner, similar to Zou [6], the performance of the suggested method in correctly selecting non-zero variables along with treating the sparsity will be evaluated. Accordingly, the performance of proposed method compared with two stated methods will be discussed in the ultra high-dimensional setting. Table 5 provides pertinent information based

**Table 4. The mean values of bias for Model 3, Model 4 and Model 5.**

| Model | n | σ | Method | | |
|---|---|---|---|---|---|
| | | | lasso | alasso | lqsso |
| Model 3 | 40 | 1 | 8.773 (0.010) | 8.766 (0.058) | **7.900** (0.887) |
| | | 3 | 10.142 (0.019) | **10.058** (0.030) | 10.161 (0.226) |
| | | 6 | 10.580 (0.011) | 10.547 (0.046) | **10.484** (0.242) |
| | 80 | 1 | 5.219 (0.126) | 5.117 (0.038) | **4.165** (0.107) |
| | | 3 | 5.553 (0.147) | 5.565 (0.045) | **5.535** (0.122) |
| | | 6 | 6.247 (0.101) | 6.257 (0.044) | **6.236** (0.203) |
| Model 4 | 40 | 1 | 1.588 (0.028) | 1.600 (0.026) | **1.524** (0.112) |
| | | 3 | 2.058 (0.008) | 2.067 (0.043) | **2.047** (0.161) |
| | | 6 | 2.661 (0.009) | **2.599** (0.061) | 2.690 (0.207) |
| | 80 | 1 | 0.514 (0.055) | 0.528 (0.027) | **0.460** (0.048) |
| | | 3 | 1.292 (0.027) | **1.010** (0.038) | 1.168 (0.040) |
| | | 6 | 1.822 (0.023) | 1.802 (0.092) | **1.684** (0.028) |
| Model 5 | 40 | 1 | 2.740 (0.004) | 2.225 (0.021) | **1.855** (0.231) |
| | | 3 | 3.766 (0.003) | 2.402 (0.008) | **1.613** (0.210) |
| | | 6 | 2.960 (0.004) | 2.651 (0.012) | **2.354** (0.482) |
| | 80 | 1 | 1.002 (0.057) | 0.795 (0.083) | **0.603** (0.371) |
| | | 3 | 1.131 (0.005) | **1.072** (0.004) | 1.153 (0.164) |
| | | 6 | 2.120 (0.005) | 2.082 (0.010) | **1.768** (0.122) |

The mean values of bias and their standard errors (in bracket), after fitting **Model 3**, **Model 4** and **Model 5** using lasso, adaptive lasso (alasso) and lqsso methods on the simulated data. For each scenario, the selected methods in each row are highlighted in bold.

upon simulation studies outlined formerly in this section. It should be pointed out that the row labeled `C` shows the number of `correctly` identified non-zero variables, and the row labeled `I` indicates the number of zero variables `incorrectly` selected by each method. Hence, the method with high and less values for `C` and `I`, respectively, is preferred.

**Table 5. Median of the number of (in)correctly selected variables for Model 5.**

| n | σ | Type | truth | Method | | |
|---|---|---|---|---|---|---|
| | | | | lasso | alasso | lqsso |
| 40 | 1 | C | 30 | 12 (0.040) | 16 (0.802) | **25** (1.727) |
| | | I | 0 | 18 (0.039) | 14 (0.764) | **5** (1.764) |
| | 3 | C | 30 | 10 (0.082) | 25 (1.140) | **28** (2.065) |
| | | I | 0 | 20 (0.081) | 4 (1.163) | **2** (2.231) |
| | 6 | C | 30 | 9 (0.077) | 21 (0.861) | **22** (2.654) |
| | | I | 0 | 21 (0.075) | 9 (0.961) | **8** (2.842) |
| 80 | 1 | C | 30 | 25 (0.710) | 27 (0.202) | **28** (2.429) |
| | | I | 0 | 5 (0710) | 3 (0.196) | **2** (2.268) |
| | 3 | C | 30 | 23 (0.258) | 23 (0.614) | **24** (1.626) |
| | | I | 0 | 7 (0.269) | 7 (0.60) | **6** (1.797) |
| | 6 | C | 30 | 19 (0.131) | 23 (0.479) | **29** (1.805) |
| | | I | 0 | 11 (0.121) | 7 (0.481) | **1** (1.824) |

Median of the number of (in)correctly selected variables and their standard errors (in bracket), after fitting **Model 5** using lasso, adaptive lasso (alasso) and lqsso methods on the simulated data for the ultra high-dimensional setting.

Recalling our simulation process, there were 30 non-zero and 970 zero coefficients while implementing **Model 5**. Accordingly, in all scenarios lqsso had a better performance in terms of correctly identifying thirty important variables compared with lasso and adaptive lasso. In addition, lqsso correctly selected non-zero variables with a relative frequency of at least 74% in the worst case. In all cases, lqsso has the least zero variables incorrectly selected during the modeling process. Indeed, those variables wrongly declaring zero, i.e. its worst case is less than 1%. Typically, a better performance of lqsso is concluded other than two alternative methods in the simulation setup.

## Real data analysis

To illustrate an application of our proposed method, we concentrate on the Bardet-Biedl syndrome gene expression data set studied by Scheetz et al. [19] and use the corresponding well-known data called the eye dataset which includes gene expression levels of $p = 18975$ genes from $n = 120$ rats. The main purpose of the analysis is to find out relevant genes that are correlated with gene TRIM32, a gene known to cause the eye disease Bardet-Biedl syndrome. Wang and Xiang [20] first screened down from 18975 genes to 3000 genes based upon the largest variances in gene expression levels. Afterwards, they computed the marginal correlation coefficients between each of these 3000 genes and the gene TRIM32, and selected the top 200 genes with the largest absolute correlation coefficients. In consequence, final data consists of 200 variables with 120 observations taken from the *flare* package [21]. We apply this final dataset, as data with $n = 120$, $p = 200$.

To conduct our analysis appropriately, we consider TRIM32 as the response variable in the proposed regression model. As discussed previously, to proceed our analysis, initial weights are not required in contrast with adaptive lasso and lqsso settings. Precisely, initial weights are set at the ridge estimates for the corresponding coefficients. The subsequent steps are designed according to the discussion represented in the paper. In other words, the main objectives are estimating the parameters by implemening different methods (lasso, adaptive lasso and lqsso) and comparing their performances with criteria provided in the simulation section.

As demonstrated in S6 Fig, the ridge estimates for coefficients, which are also considered as initial coefficients based upon the suggestion made by Zou [6], vary between -0.1 to 0.1 in magnitude. It is remarkable that estimates have a clear sign of conformity with the discussion presented for **Model 4** and **Model 5** in our simulation studies. Although the numbers of variables applied here is less, it might be claimed that this example is mostly relevant to **Model 5**. Hence, while fitting a regression model to analyze our real data example, we initially rely on this model to treat small effects scenario. In consequence, we expect that lqsso will provide a better accuracy and precision in capturing the variability in this data set.

For the comparison purpose, we compute the $n$-fold cross-validation test error, abbreviated as CVErr, i.e.

$$CV\ Err = mean\{(y_1 - y_{-1,p})^2, \ldots, (y_n - y_{-n,p})^2\}, \tag{9}$$

where $y_i$ is $i$-th observation of the response variable and $y_{-i,\ p}$ refers to the predicted value by fitting a model using all observations except $i$-th sample. The latter technique is also nominated as leave-one-out cross-validation (sometimes LOOCV in abbreviation).

As demonstrated in Table 6, various methods function differently from each other. Note that the last lines at the end of table exhibit the accuracy criterion. As indicated, the inserted methods represent sparse solution in all covariates except in gene numbered 21094, 22016, 23041, 24565 and 29842. Therefore, the stated variables play an important role in causing eye disease, usually assigned as substantial and significant variables in this particular example. As a

**Table 6. Estimated mean values for the coefficients in rat eye dataset.**

| Predictor | lasso | alasso | lqsso |
|---|---|---|---|
| **21094** | 0.139 | 0.165 | 0.139 |
| **22016** | 0.172 | 0.184 | 0.172 |
| **23041** | 0.618 | 0.609 | 0.618 |
| **24565** | 0.054 | 0.045 | 0.054 |
| **29842** | 0.044 | 0.032 | 0.044 |
| **CVErr** | 0.013 (0.0004) | 0.015 (0.003) | **0.011** (0.0002) |

The estimated mean values for cross-validation test error, their standard errors (in bracket) and the coefficients using lasso, adaptive lasso (alasso) and lqsso methods along with the model selection criteria in analyzing the rat eye data.

result, lqsso has the best performance in terms of CVErr criteria. Moreover according to the results, the superiority of suggested penalization method is apparent in comparison with previous lasso techniques. Another worth mentioning subject is the magnitude of estimated values. The lasso and lqsso provided the same estimates but CVErr regarding to lqsso is lower. In conclusion in this particular example, it should be noted that estimated values corresponding to $\lambda_n$, $\gamma$ and $\tau$ were 0.4, 0.1 and 0.63, respectively.

## Conclusion

In this article, we defined a novel method in the structure of penalized regression problem. The proposed method is under the same umbrella as the renowned approach lasso does. The suggested method, denominated as lqsso, is able to treat unusual observations as well as selecting important variables. Moreover, the suggested method can appropriately deal with sparsity in various dimensional problems, i.e. low, moderate, high and ultra high-dimensional. Simulation studies conducted in this paper reveal the superiority of our proposed method in contrast with lasso and adaptive lasso in several small effect situations. Our claim is effectively confirmed regarding to RPEs and their standard errors in various simulation scenarios, particularly in ultra high-dimensional setting. Additionally, while analyzing a real data set, our proposed method has proved better performance compared with alternative methods.

We illustrated that our proposed method enjoys oracle properties under some mild conditions. In this connection, lqsso reacts the same as adaptive lasso does. Our proposed method provides lower bias than adaptive lasso, but not as good as lasso. Nevertheless, our suggested method performs well in the context of variable selection and sparsity compared with lasso. Considering RPE measure, our proposed method has a prime performance in comparison with two alternative methods, particularly in an ultra high-dimensional setting. As Zou [6] pointed out, there is no superior method in all situations, regarding what we mentioned in this article.

Similar to lasso, our proposed method can also be figured out in a Bayesian methodology. However, adaptive lasso can not be considered in this framework. The lqsso method can be remarked as a Maximum A Posteriori (MAP) estimator, similar to ridge and lasso regression. See, for instance, [22]. This will be true if the prior distributions for coefficients is considered as the Asymmetric Laplace Distribution (ALD). One can consult [10] for more details on this latter subject. We aim to develop such viewpoint in our future research.

The construction of quantile regression mainly arises from considering check function, a robust measure for treating outliers. Therefore, quantile regression is a prime trick to deal with unusual coefficients while considering a penalty function in a minimizing scheme. It is

important to state that, with some prior knowledge on the skewness of regression coefficients, there is possibility and background to scrutinize an optimization algorithm for choosing appropriate weights. This topic will be our effort for further research.

## Supporting information

**S1 Fig. Plot of the geometry of the lqsso with different $\tau$.** Plot of the lqsso estimators with $\tau$ = .2, $\tau$ = .5 (approximately equal to the lasso) and $\tau$ = .8 sketched at the left, middle and right panels, respectively, centered at OLS estimates. See the text for more details.
(TIFF)

**S2 Fig. Plot of a comparison among the lqsso, lasso, and adaptive lasso.** Plot of the lqsso penalty in comparison with the lasso and adaptive lasso penalty according to the plot C in Zou [6].
(TIFF)

**S3 Fig. Plot of the Bayesian view of the lqsso penalty.** Plot of the lqsso penalty in comparison with the lasso and adaptive lasso penalty based on Bayesian approach.
(TIFF)

**S4 Fig. The RPEs comparison of the three methods.** Compare methods in terms of the RPE criterion using different models, sample sizes ($n$) and the standard deviation ($sd$) of the error term.
(PNG)

**S5 Fig. The biases comparison of the three methods.** To compare methods in terms of the bias criterion using different models, sample sizes ($n$) and the standard deviation ($sd$) of the error term.
(TIFF)

**S6 Fig. The initial ridge coefficients.** The plot shows the ridge estimates for the coefficients of the corresponding ridge regression model.
(TIFF)

## Acknowledgments

Receiving support from the Center of Excellence in Analysis of Spatio-Temporal Correlated Data at Tarbiat Modares University is acknowledged.

## Author Contributions

**Formal analysis:** Alireza Daneshvar.

**Project administration:** Golalizadeh Mousa.

**Supervision:** Golalizadeh Mousa.

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
