## [Decision Letter · Decision Letter 0]

13 Oct 2021

PONE-D-21-27670Regression Shrinkage and Selection via Least Quantile Shrinkage and Selection OperatorPLOS ONE

Dear Dr. Golalizadeh,

Thank you for submitting your manuscript to PLOS ONE. After careful consideration, we feel that it has merit but does not fully meet PLOS ONE’s publication criteria as it currently stands. Therefore, we invite you to submit a revised version of the manuscript that addresses the points raised during the review process.

We look forward to receiving your revised manuscript.

Kind regards,

Xiaoyu Song

Academic Editor

PLOS ONE

Journal Requirements:

3. Please ensure that you refer to Figures 3 and 4 in your text as, if accepted, production will need this reference to link the reader to the figure.

4. Please remove your figures from within your manuscript file, leaving only the individual TIFF/EPS image files, uploaded separately. These will be automatically included in the reviewers’ PDF.

Additional Editor Comments:

The authors proposed an interesting idea to extend LASSO to have a quantile based weight. The rationals for using quantile based weights are not fully introduced. How quantile based weights can be linked with the quantile of outcomes, and why they are linked with the directions of the coefficients are not well explained. The establishment of statistical properties (e.g. oracle property) needs further explanation, and simulations should report more scenarios and/or model results to demonstrate the performance of the proposed method.

Reviewers' comments:

Reviewer's Responses to Questions

**Comments to the Author**

1. Is the manuscript technically sound, and do the data support the conclusions?

Reviewer #1: Yes

Reviewer #2: Partly

2. Has the statistical analysis been performed appropriately and rigorously? 

Reviewer #1: Yes

Reviewer #2: Yes

3. Have the authors made all data underlying the findings in their manuscript fully available?

Reviewer #1: Yes

Reviewer #2: Yes

4. Is the manuscript presented in an intelligible fashion and written in standard English?

Reviewer #1: Yes

Reviewer #2: No

5. Review Comments to the Author

Reviewer #1: The manuscript introduced a new method in the class of the penalized regression approach. It is a solid contribution to the field but some important pieces in the simulation result section are missing. I do believe, however, a substantial revision will help make it a more valuable contribution to the literature.

Major comments:

1. In the simulation studies section, the authors compared RPEs in Table 1 and 4, bias in Table 2 and 4, number of (in)correct selected variables in Table 5. However, the authors only provided the standard errors for RPEs in Table 1 and 4. I am wondering whether the authors could also provide standard errors for bias and number of (in)correct selected variables in other tables.

2. For number of (in)correct selected variables in Table 5, I would recommend using the mean and standard errors instead of median, just to be consistent with the previous 4 tables. Are there any reasons to use median here?

3. Similarly, in Table 6, please provide the standard error for the cross validation so that audience can know the variability of the metric.

Minor comments:

1. The 3rd paragraph in the Introduction section (From "One of the existing" to "for the coefficients") talks about the potential Bayesian understanding for the new approach . I would recommend moving it to the Discussion section or cut it shorter as it is beyond the function of the "Introduction" section.

2. I would strongly recommend the authors to make the code of new method publicly available so that people from a broader community can utilize.

3. The figures need to be improved. For example, in Fig 1, the scale has been distorted. In Fig 2, please get rid of the "_" in the figure labels.

4. In Table 1 to Table 6, I think the authors are providing the mean values. Please specify that instead of just putting "values" and let the readers turn back and forth to find out what the values mean.

Reviewer #2: Instead of using the originally proposed weight: w_j = 1 / |\\beta_j|^\\gamma in adaptive LASSO, the authors proposed to use w_j = (1-\\tau)*I(\\beta_j <= 0) + \\tau*I(\\beta_j > 0) as the new weight. The weight takes either \\tau or 1-\\tau depending on the sign of initial coefficient. Since the weight is between 0 and 1, the following problem is solved: if the absolute of the initial coefficient is less than 1, the weight will be large leading to increase in bias. The authors did extensive simulation studies and a real study, and showed that the proposed LQSSO outperforms LASSO and adaptive LASSO in terms of Relative Prediction Error of Y and estimation bias of \\beta. The work is well-organized and the numerical studies are informative. Comparing to the work of Zou (2006), I would like to see the followings addressed:

1. When \\tau=0.5, LQSSO is equivalent to LASSO. Since LASSO doesn't have oracle property in general, how could the oracle property of LQSSO be established? Is there some non-trivial necessary condition missing?

2. Why is the proposed weight superior? Or break down to (1) Why does considering the sign of initial coefficients help? e.g., Given \\tau=0.2, w=0.8 when the initial coefficient (usually OLS or Ridge estimates) is negative and w=0.2 when it is positive. (2) If the goal is to constrain the weight between 0 and 1, how about sth like w_j = 1 / (1 + exp(|\\beta|))?

3. Please provide literature or numerical or theoretical support for the claim: If the absolute of the initial coefficient is less than 1, the weight will be large leading to increase in bias.

4. Page 6: ''This is in contrast with other penalty proposed so far in which the response's quantile did not play a role in the penalty.'' So, how does LQSSO include Y's quantile in the penalty with w_j = (1-\\tau)*I(\\beta_j <= 0) + \\tau*I(\\beta_j > 0)?

5. Fig 1 is a nice illustration, while (1) Given one \\tau, suppose it is the \\tau selected after tuning, the constraint region is still a trapezoid, why does it provide more comprehensive insight than LASSO? (2) Please also plot a figure similar to Fig 1(c) of Zou (2006). I guess it is a horizontal shift of Fig 1(c). So this is related to Question 2, what does LQSSO gain or lose from the shift?

6. Please check statements and language, some are confusing. e.g., Page 7, last paragraph: ''when the initial weights for the coefficients in the adaptive LASSO are less than one'', what does it refer to, w_j or \\beta_j?

6. PLOS authors have the option to publish the peer review history of their article (what does this mean?). If published, this will include your full peer review and any attached files.

Reviewer #1: No

Reviewer #2: No

---

## [Author Response · Author response to Decision Letter 0]

2 Nov 2021

Responses to Reviewer #1

Major comments:

1) In the simulation studies section, the authors compared RPEs in Table 1 and 4, bias in Table 2 and 4, number of (in)correct selected variables in Table 5. However, the authors only provided the standard errors for RPEs in Table 1 and 4. I am wondering whether the authors could also provide standard errors for bias and number of (in)correct selected variables in other tables.

The standard errors have been reported for the biases and the number of (in)correct selected variables and they are added in the relevant tables.

2) For number of (in)correct selected variables in Table 5, I would recommend using the mean and standard errors instead of median, just to be consistent with the previous 4 tables. Are there any reasons to use median here?

Following Zou (2006), because the true numbers are inherently integer values, we prefer to report the median for the number of (in)correct selected variables. Moreover, as suggested, the standard errors of those numbers were also reported in the related tables.

3) Similarly, in Table 6, please provide the standard error for the cross validation so that audience can know the variability of the metric

The standard errors for the cross-validations were also reported in Table 6. We appreciate your suggesting this point.

Minor comments:

1) The 3rd paragraph in the Introduction section (From "One of the existing" to "for the coefficients") talks about the potential Bayesian understanding for the new approach. I would recommend moving it to the Discussion section or cut it shorter as it is beyond the function of the "Introduction" section.

Our Bayesian view of proposed idea became shorter in the Introduction, and more comments are added in the Conclusion section.

2) I would strongly recommend the authors to make the code of new method publicly available so that people from a broader community can utilize.

Thanks for this recommendation. However, we currently cannot share our code publicly because we’re still working on it for other research studies. Notably, we are improving the idea of quantile regression and mixed quantile regression with lqsso penalty.

3) The figures need to be improved. For example, in Fig 1, the scale has been distorted. In Fig 2, please get rid of the "_" in the figure labels.

The scales set in figure 1. Also, we corrected the order of \\tau=0.2, 0.5 and 0.8 in this figure. Also, we removed the “_” sign in the Y label of the current version of figure 2. 

4) In Table 1 to Table 6, I think the authors are providing the mean values. Please specify that instead of just putting "values" and let the readers turn back and forth to find out what the values mean.

Those were good points. Now, we replaced the “values” by the “mean values” in Table 1 to Table 6 except in Table 5, which is, in fact, the median. 

Responses to Reviewer #2

1) When \\tau=0.5, LQSSO is equivalent to LASSO. Since LASSO doesn't have oracle property in general, how could the oracle property of LQSSO be established? Is there some non-trivial necessary condition missing? 

That is true that the Lqsso with \\tau=0.5 is equivalent to Lasso graphically but this is not in reality. The reason is that the lqsso is approximately equal to the lasso. Specifically, the main difference between the two is that lqsso consists of the indicator function of OLS coefficients in the penalty component, making it different from LASSO if \\tau=0.5.

2) Why is the proposed weight superior? Or break down to (1), Why does considering the sign of initial coefficients help? e.g., Given \\tau=0.2, w=0.8 when the initial coefficient (usually OLS or Ridge estimates) is negative and w=0.2 when it is positive. (2) If the goal is to constrain the weight between 0 and 1, how about sth like w_j = 1 / (1 + exp(|\\beta|))?

Compared with your proposed penalty w_j = 1 / (1 + exp(|\\beta|)), lqsso relies on check function (\\rho _{\\tau}(.)). So it has closed-form and many representations to be used in various circumstances such as implementing an algorithm, invoking in a geometrical view, recalling a Bayesian view, using for proof of Oracle property, and calculating KKT conditions (closed form) in the quantile regression with lqsso penalty. Other representations are as follows.

The signs show the direction of the polygon in the figure and OLS coefficients in the penalty part. More precisely, OLS coefficients are based on a density function f(y|X), so it considers the mass not only at the middle of the density but also at their two tails, especially when f(.) is asymmetric. Hence it is more flexible more than lasso and adaptive.

Plot C in Zou (2006) is as follows for lqsso (black points). Note that Zou (2006) indicated them by a line in his graph, but we preferred to plot them as points for better illustration purposes. Ali, it is not clear and lasso (Green line). It shows according to the \\tau value and its sign, lqsso line how much is closer to the red line and also on the positive or negative side.

Moreover, one can use the ALD as a prior distribution for the initial coefficients and obtain the estimations of the lqsso via computing the MAP estimator. This is the case if one considers the ALD as the prior distributions for the initial coefficients. Johnstone and Titterington (2009) also presented a Bayesian view to the lasso model by setting up the prior density for the coefficients as the Standard Laplace distribution. To do so, the zero coefficients will have a high probability compared to the non-zeros, and this feature would guarantee the sparsity. The plot given below shows this exciting property. To the best of our knowledge, the adaptive lasso has not been treated from a Bayesian perspective. We expect this gap can be filled via considering a re-scaled Standard Laplace distribution. This topic will be tackled in our future research.

Let us go back to your question, i.e., the effect of the signs. As mentioned, we should initially fix \\tau, e.g., via invoking the cross-validation technique. Now, assume that \\tau=0.8, where \\beta_1>0 and less than 1 (consequently w_1>1). Then, we could assign a probability for this true coefficient and a reasonable probability for zero coefficients, i.e., sparsity. This idea owns to the signs.

Moreover, our weights will still be between 0 and 1. Note that it advocates the highest probability for zero for the adaptive case, as seen in the plot. 

As a summary, the table below indicates where our proposed method shows superiority compared with other methods.

Property Lasso Adaptive Lqsso

Oracle Yes Yes

Bayesian view and rationale Yes Yes

Use lasso algorithm Yes Yes Yes

Weights are based on random variables & observations Yes Yes

Geometry aspect Yes Yes

3) Please provide literature or numerical or theoretical support for the claim: If the absolute of the initial coefficient is less than 1, the weight will be large leading to increase in bias. 

The rationale behind this claim includes the following sequential statements:

a) If the absolute of the initial coefficient is less than 1, the weight will be large.

b) In a regular regression (OLS), there is no penalty part, bias is low and variance is high. By using a penalty, we sacrifice bias (has increased) to reduce variance i.e., bias-variance tradeoff. Hence, with adding a penalty, bias increases.

c) According to Buhlmann and Van De Geer (2011), if |\\beta_ j,initial| is large, the adaptive Lasso employs a minor penalty (i.e. little shrinkage) for the jth coefficient (b_j) which implies less bias. 

d) The topic has not already been appropriately addressed in the literature. Albeit, Zou (2006), B{\"u}hlmann and Van De Geer (2011) and Fan et al. (2009) suggested some easy solutions such as taking the initials coefficients equal to zero if that particular circumstance occurs.

So, considering a&b or (c&d, with less weights) support our claim. 

4) Page 6: ''This is in contrast with other penalty proposed so far in which the response's quantile did not play a role in the penalty.'' So, how does LQSSO include Y's quantile in the penalty with w_j = (1-\\tau)*I(\\beta_j <= 0) + \\tau*I(\\beta_j > 0)?

We should point out that our proposed idea is not directly based on response's quantile. But, generally, the regression modeling set-up is related to the quantity E(y|X,\\beta,\\tau) which takes the randomness of Y into account. Moreover, we do see that the penalty term utilized the estimate of the OLS coefficients and these latter values are derived via the density function f(y|X), or directly speaking the randomness of Y. Hence, our considered check function, i.e. \\rho, inherits Y’s quantile.

5) Fig 1 is a nice illustration, while (1) Given one \\tau, suppose it is the \\tau selected after tuning, the constraint region is still a trapezoid, why does it provide more comprehensive insight than LASSO? (2) Please also plot a figure similar to Fig 1(c) of Zou (2006). I guess it is a horizontal shift of Fig 1(c). So this is related to Question 2, what does LQSSO gain or lose from the shift?

The answer to this comment is provided in replying to the comment 2.

6) Please check statements and language, some are confusing. e.g., Page 7, last paragraph: ''when the initial weights for the coefficients in the adaptive LASSO are less than one'', what does it refer to, w_j or \\beta_j?

The correct sentence is “when the initial coefficients for the weights in the adaptive lasso are less than one.” We corrected this in the revised version.

The best regards

Mousa

---

## [Decision Letter · Decision Letter 1]

26 Dec 2021

PONE-D-21-27670R1Regression Shrinkage and Selection via Least Quantile Shrinkage and Selection OperatorPLOS ONE

Dear Dr. Golalizadeh,

Thank you for submitting your manuscript to PLOS ONE. After careful consideration, we feel that it has merit but does not fully meet PLOS ONE’s publication criteria as it currently stands. Therefore, we invite you to submit a revised version of the manuscript that addresses the points raised during the review process.

Some of the responses provided in this revision are not well organized, vaguely stated, and hard to understand. None of the important points (Reviewer 2 Questions 1-4) are addressed in the manuscript. I also suggest they have a professional editing of the language, making the statements clear and coherent.

We look forward to receiving your revised manuscript.

Kind regards,

Xiaoyu Song

Academic Editor

PLOS ONE

Journal Requirements:

Additional Editor Comments (if provided):

Reviewers' comments:

Reviewer's Responses to Questions

**Comments to the Author**

1. If the authors have adequately addressed your comments raised in a previous round of review and you feel that this manuscript is now acceptable for publication, you may indicate that here to bypass the “Comments to the Author” section, enter your conflict of interest statement in the “Confidential to Editor” section, and submit your "Accept" recommendation.

Reviewer #1: All comments have been addressed

Reviewer #2: (No Response)

2. Is the manuscript technically sound, and do the data support the conclusions?

Reviewer #1: Yes

Reviewer #2: Partly

3. Has the statistical analysis been performed appropriately and rigorously? 

Reviewer #1: Yes

Reviewer #2: Yes

4. Have the authors made all data underlying the findings in their manuscript fully available?

Reviewer #1: Yes

Reviewer #2: Yes

5. Is the manuscript presented in an intelligible fashion and written in standard English?

Reviewer #1: Yes

Reviewer #2: No

6. Review Comments to the Author

Reviewer #1: (No Response)

Reviewer #2: Thanks for the explanations in the response. Overall, it would be better if (1) The explanations in response could be more organized and polished, e.g., no need to include discussions between authors, and please explain everything clearly and to the very point (especially for Answers 1-3). (2) Most of the explanations are informative and convincing, thus could be addressed or incorporated in the main text or supplement, so that readers like me will be convinced.

7. PLOS authors have the option to publish the peer review history of their article (what does this mean?). If published, this will include your full peer review and any attached files.

Reviewer #1: No

Reviewer #2: No

---

## [Author Response · Author response to Decision Letter 1]

8 Feb 2022

As we understood, the Reviewer 1is fully satisfied with the changes made in our first revised paper as well as the responses to her/his comments. The Reviewer 2, but, was mostly concerned about the questions (we assume those were default of the PlOS-One) 2 and 6 in your previous email. She (he) was partly happy with the question 2 “2. Is the manuscript technically sound, and do the data support the conclusions?”. Hence, we attempted to improve the paper to fit with this request. Considering the question 6 “Review Comments to the Author “, we tried to fit ourselves with her/his request in the points (1) and (2) and set all necessary materials in the paper, particularly where the concepts should have been cleared. We hope the changes made in those relevant pages, sections and expressions are well enough to convince her/him.

---

## [Decision Letter · Decision Letter 2]

18 Mar 2022

Regression Shrinkage and Selection via Least Quantile Shrinkage and Selection Operator

PONE-D-21-27670R2

Dear Dr. Golalizadeh,  

We’re pleased to inform you that your manuscript has been judged scientifically suitable for publication and will be formally accepted for publication once it meets all outstanding technical requirements.

Kind regards,

Xiaoyu Song

Academic Editor

PLOS ONE

Additional Editor Comments (optional):

Reviewers' comments:

Reviewer's Responses to Questions

**Comments to the Author**

1. If the authors have adequately addressed your comments raised in a previous round of review and you feel that this manuscript is now acceptable for publication, you may indicate that here to bypass the “Comments to the Author” section, enter your conflict of interest statement in the “Confidential to Editor” section, and submit your "Accept" recommendation.

Reviewer #2: All comments have been addressed

2. Is the manuscript technically sound, and do the data support the conclusions?

Reviewer #2: Yes

3. Has the statistical analysis been performed appropriately and rigorously? 

Reviewer #2: Yes

4. Have the authors made all data underlying the findings in their manuscript fully available?

Reviewer #2: Yes

5. Is the manuscript presented in an intelligible fashion and written in standard English?

Reviewer #2: Yes

6. Review Comments to the Author

Reviewer #2: (No Response)

7. PLOS authors have the option to publish the peer review history of their article (what does this mean?). If published, this will include your full peer review and any attached files.

Reviewer #2: No

---

## [Editor Report · Acceptance letter]

29 Sep 2022

PONE-D-21-27670R2 

Regression shrinkage and selection via least quantile shrinkage and selection operator 

Dear Dr. Golalizadeh:

I'm pleased to inform you that your manuscript has been deemed suitable for publication in PLOS ONE. Congratulations! Your manuscript is now with our production department. 

Kind regards, 

on behalf of

Dr. Xiaoyu Song 

Academic Editor

PLOS ONE